# The Roles of AGTRAP, ALKBH3, DIVERSIN, NEDD8 and RRM1 in Glioblastoma Pathophysiology and Prognosis

**DOI:** 10.3390/biomedicines12040926

**Published:** 2024-04-22

**Authors:** Claudia Alexandra Dumitru, Nikolas Walter, Carl Ludwig Raven Siebert, Frederik Till Alexander Schäfer, Ali Rashidi, Belal Neyazi, Klaus-Peter Stein, Christian Mawrin, Ibrahim Erol Sandalcioglu

**Affiliations:** 1Department of Neurosurgery, Otto-von-Guericke University, 39120 Magdeburg, Germanyerol.sandalcioglu@med.ovgu.de (I.E.S.); 2Department of Neuropathology, Otto-von-Guericke University, 39120 Magdeburg, Germany

**Keywords:** IDH wild-type glioblastoma, biomarkers, overall survival, progression-free survival, prognostic accuracy, marker combinations

## Abstract

This study determined the expression of five novel biomarker candidates in IDH wild-type glioblastoma (GBM) tissues compared to non-malign brain parenchyma, as well as their prognostic relevance for the GBM patients’ outcomes. The markers were analysed by immunohistochemistry in tumour tissues (n = 186) and healthy brain tissues (n = 54). The association with the patients’ overall survival (OS) and progression-free survival (PFS) was assessed by Kaplan–Meier and log-rank test. The prognostic value of the markers was determined using multivariate Cox proportional hazard models. AGTRAP, DIVERSIN, cytoplasmic NEDD8 (NEDD8c) and RRM1 were significantly overexpressed in tumour tissues compared to the healthy brain, while the opposite was observed for ALKBH3. AGTRAP, ALKBH3, NEDD8c and RRM1 were significantly associated with OS in univariate analysis. AGTRAP and RRM1 were also independent prognostic factors for OS in multivariate analysis. For PFS, only AGTRAP and NEDD8c reached significance in univariate analysis. Additionally, AGTRAP was an independent prognostic factor for PFS in multivariate models. Finally, combined analysis of the markers enhanced their prognostic accuracy. The combination AGTRAP/ALKBH3 had the strongest prognostic value for the OS of GBM patients. These findings contribute to a better understanding of the GBM pathophysiology and may help identify novel therapeutic targets in this type of cancer.

## 1. Introduction

Glioblastoma (GBM) is the most prevalent malign brain tumour in adult patients and one of the most deadly types of cancer overall, with 5-year survival rates of less than 7% [1,2]. Since the newest classification from 2021, only the WHO grade 4 diffuse gliomas that occur in adults and have an IDH wild-type phenotype are considered as ‘true’ glioblastomas [3]. Because of its highly infiltrative growth, this tumour cannot be completely removed by surgery and necessitates a multimodal therapeutic approach. However, after the development of the ‘Stupp protocol’ almost 20 years ago [4], there have been no other major breakthroughs in the therapy of GBM. The focus of recent research has therefore shifted towards understanding the pathophysiology of GBM, in order to develop specific targeted therapies. To reach this goal, it is necessary to identify novel cellular and molecular factors that control GBM progression and could serve as therapeutic targets in this type of cancer. An important first step in this process is the identification of biomarkers which are differentially expressed in GBM compared to the healthy brain tissue and/or associate with the patients’ outcome. 

In this study, we investigated five potential biomarker candidates that have been linked to tumour progression and prognosis in several types of cancer. AGTRAP (Angiotensin II Type I Receptor-associated Protein) is a member of the renin–angiotensin system, which specifically interacts with the AT1 receptor to regulate cardiovascular and fluid homeostasis [5]. In cancer, AGTRAP can promote tumour progression through activation of the MAPK and AKT/mTOR signaling pathways [6,7], but potentially also through regulation of immunological processes, such as T-cell exhaustion or recruitment of pro-inflammatory and immunosuppressive immune cells [6,8]. ALKBH3 (AlkB Homolog 3)/PCA-1 (Prostate Cancer Antigen-1) is a DNA/RNA repair enzyme, which acts as a demethylase to regulate cellular metabolism (reviewed in [9]). Currently, there is no consensus regarding the role of ALKBH3 in cancer. While some studies found that ALKBH3 was overexpressed in tumour tissues and exhibited tumour-promoting functions [10,11,12], others showed that ALKBH3 was epigenetically inactivated in certain malignancies. The latter phenomenon associated with a poor outcome of the respective cancer patients [13,14,15]. DIVERSIN, also known as ANKRD6 (Ankyrin Repeat Domain-containing Protein 6), is a modulator of the Wnt and Wnt/JNK signaling pathways [16]. Thus far, most studies consider DIVERSIN to be a pro-tumour factor, as it enhances tumour proliferation and invasion in a variety of malignancies, such as non-small cell lung cancer, colorectal carcinoma, breast cancer and glioma [17,18,19,20]. Additionally, recent studies linked this marker to immunological processes in colon cancer, where DIVERSIN expression correlated positively with the numbers of tumour-promoting M2 macrophages and negatively with the numbers of anti-tumour effector cells [21]. NEDD8 (Neuronal precursor cell-Expressed Developmentally Down-regulated protein 8) is a ubiquitin-like molecule, which conjugates to a lysine residue of the substrate protein [22]. This mechanism called ‘neddylation’ can promote—when overactivated—the degradation of tumour suppressor proteins, such as p21 and p27, thereby facilitating carcinogenesis and tumour progression (reviewed in [23]). Importantly, neddylation can be inhibited by the FDA-approved drug MLN4924/pevonedistat, which makes it a promising target for cancer therapy (reviewed in [24]). RRM1 (Ribonucleotide Reductase large subunit M1) is the catalytic subunit of the ribonucleotide reductase [25]. Similar to ALKBH3, there have been contradictory reports in the literature regarding the role of RRM1 in cancer [26,27,28,29], reviewed in [30]. In glioma however, RRM1 appears to be a tumour-promoting factor, as RRM1 silencing induced cell cycle arrest and inhibited the proliferation of the tumour cells [31]. Furthermore, very recent studies linked high levels of RRM1 to the poor outcome of GBM patients and identified RRM1 as a promising target for drug combination therapy in this type of cancer [32].

The role of these markers in GBM pathophysiology and prognosis is insufficiently characterized thus far. In particular, studies on homogenous cohorts of IDH wild-type GBM patients employing multivariate proportional hazard models are largely missing. The purpose of our study was to assess (1) the expression of the markers in IDH wild-type GBM tissues compared to non-malign adjacent brain parenchyma, (2) the association between marker expression and the outcome of the patients, (3) the prognostic value of the markers for the patients’ outcome using multivariate models and (4) the prognostic accuracy of the markers analysed individually or in combination.

## 2. Materials and Methods

### 2.1. Study Subjects

This study was conducted on adult patients with newly diagnosed glioblastoma (GBM). Only the patients with confirmed IDH wild-type status were included in the analysis (n = 186). The median age of the patients was 66 years. All patients were treated at the Department of Neurosurgery, University Hospital Magdeburg between 2008 and 2018. The study was conducted in accordance with the Declaration of Helsinki issued in 1975 and revised in 2013. The ethics committee of the Otto-von-Guericke University Magdeburg approved the study (No. 146/2019) and waivered the need for informed written consent. The main clinical characteristics of the GBM patients are summarized in Table 1.

### 2.2. Marker Expression and Scoring

The expression of the biomarker candidates was assessed in FFPE (formalin-fixed paraffin-embedded) tissues, which were processed as tissue microarrays (TMAs) as in previous studies [33,34]. Both tumour tissues (n = 186) and non-malign brain parenchymas (n = 54) were included in the analysis. The latter consisted of brain tissue distant from the solid tumour area in order to minimize the effect of the infiltrating/diffuse-growing tumour cells on the subsequent analysis. The samples were stained with the following primary antibodies: 193 ng/mL polyclonal rabbit anti-AGTRAP, 293 ng/mL polyclonal rabbit anti-ALKBH3, 333 ng/mL polyclonal rabbit anti-ANKRD6/DIVERSIN, 114 ng/mL monoclonal rabbit anti-NEDD8 (19E3) and 248 ng/mL monoclonal rabbit anti-RRM1 (D12F12). The AGTRAP, ALKBH3 and DIVERSIN antibodies were purchased from Proteintech Europe (Manchester, UK), while NEDD8 and RRM1 were from Cell Signaling Technology (Frankfurt am Main, Germany). After staining, the samples were digitalized with an Aperio VERSA high-resolution whole slide scanner and analysed with the Aperio ImageScope V12.1.0.5029 software (both from Leica Biosystems, Nussloch, Germany). Authors C.A.D., N.W., C.L.R.S. and F.T.A.S. performed blinded histological scoring independently.

AGTRAP, ALKBH3, DIVERSIN and RRM1 displayed a cytoplasmic subcellular localization. NEDD8 displayed both a cytoplasmic (NEDD8c) and a nuclear (NEDD8n) subcellular localization. According to the intensity, we categorized the cytoplasmic staining as ‘weak’ (1 point), ‘medium’ (2 points) and ‘strong’ (3 points) (Figure 1A–D,F). To account for potential heterogeneity in the staining pattern among samples, we assessed the final expression levels of the cytoplasmic markers using the H-Score: (1 × X) + (2 × Y) + (3 × Z), where X + Y + Z = 100% of the total tumour area. For NEDD8n, we assessed the percentage of positive cells using a 5-tier scoring system: <20%, 20–40%, 41–60%, 61–80% and >80%, and we assigned 0, 1, 2, 3 and 4 points, respectively (Figure 1E). A high-magnification micrograph showing both positive and negative NEDD8n tumour cells is presented in Appendix A. At least three different fields per TMA spot were analysed at 200-fold magnification, and the values were subsequently averaged. 

### 2.3. Statistical Analysis

The data were analysed with the SPSS version 29.0.1.0 (171) software (IBM Corporation, Armonk, NY, USA). The difference in marker expression between tumour and non-malign brain parenchyma was analysed using box-whisker plots. For this set of studies, the statistical significance was assessed with the Mann–Whitney U-test. Survival curves (36 months for overall survival and 12 months for progression-free survival) were generated with the Kaplan–Meier method. The significance was tested both by univariate analysis using the log-rank test and by multivariate Cox proportional hazard regression models adjusted for age, Karnofsky performance scale, extent of surgical resection and MGMT methylation status.

## 3. Results

### 3.1. Marker Expression in Tumour versus Healthy Brain Tissues

In the first set of studies, we sought to determine whether our biomarker candidates were differentially expressed in glioblastoma (GBM) compared to the adjacent non-malign brain tissue. The results showed that the levels of AGTRAP (Figure 2A,B), DIVERSIN (Figure 2E,F), NEDD8c (Figure 2G,H) and RRM1 (Figure 2I,J) were significantly higher in the tumour tissues compared to the healthy brain parenchyma. In contrast, ALKBH3 was significantly downregulated in GBM compared to the healthy brain (Figure 2C,D). The expression of NEDD8n was not significantly different between the two groups of samples (*p* = 0.737, Mann–Whitney U). These results were confirmed when the group sizes were adjusted to include only the cases where both healthy and tumour tissues were available (n = 54) (Appendix A).

### 3.2. Marker Expression and the Overall Survival of GBM Patients

Next, we tested whether our biomarker candidates might associate with the overall survival (OS) of the GBM patients. To this end, we first dichotomised the expression levels of the markers into ‘low’ and ‘high’ according to the median-split method. We subsequently plotted 36-month Kaplan–Meier survival curves for each marker and performed univariate survival analyses using the log-rank test. The results showed that patients with high levels of AGTRAP had a significantly shorter OS compared to patients with low AGTRAP levels (*p* < 0.001, log-rank; Figure 3A). Similarly, high levels of NEDD8c were significantly associated with a shorter OS of the GBM patients (*p* = 0.03, log-rank; Figure 3D). ALKBH3 was significantly associated with the OS of the GBM patients as well, but in this case, the patients with high levels of ALKBH3 survived significantly better than the ALKBH3^low^ patients (*p* = 0.039, log-rank; Figure 3B).

For DIVERSIN (Figure 3C), NEDD8n (Figure 3E) and RRM1 (Figure 3F), there was no significant difference regarding OS between patients with high and those with low expression levels of these markers (*p* = 0.232, *p* = 0.753 and *p* = 0.296, log-rank, respectively). We consequently sought to determine whether the dichotomisation of these markers into ‘low’ and ‘high’ using a new cut-off might yield other results. Since DIVERSIN and RRM1 were differentially expressed in healthy brain compared to the GBM tissues, we selected the median values of the healthy samples as the new cut-off. Even under these conditions, there was no significant difference in OS between DIVERSIN^high^ and DIVERSIN^low^ patients (*p* = 0.929, log-rank; Figure 4A). However, we now found a significant association between high levels of RRM1 and a shorter survival of the GBM patients (*p* = 0.031, log-rank; Figure 4B). NEDD8n could not be analysed with the new cut-off, as its expression levels were similar in both healthy and tumour tissues.

In further studies, we performed multivariate survival analyses using Cox proportional hazard models adjusted for potential confounders, such as age, Karnofsky Performance Scale (KPS), extent of surgical resection and MGMT methylation status [35,36,37,38]. We only carried out these tests for the markers that were significantly associated with OS in the univariate analysis. The multivariate models showed that high levels of AGTRAP and of RRM1 significantly predicted the shorter OS of GBM patients (AGTRAP: HR = 1.814, 95% CI = 1.260–2.612, *p* = 0.001; RRM1: HR = 1.746, 95% CI = 1.099–2.775, *p* = 0.018). In contrast, ALKBH3 and NEDD8c did not reach significant prognostic values regarding the OS of these patients. The results of the multivariate analysis are summarized in Table 2.

### 3.3. Marker Expression and the Progression-Free Survival of GBM Patients

Next, we tested whether our biomarker candidates might associate with the 12-month progression-free survival (PFS) of the GBM patients. The survival analysis was performed as described above. From all markers tested, only two showed significant associations with the PFS of GBM patients. Specifically, patients with high levels of AGTRAP (Figure 5A) and NEDD8c (Figure 5D) had a significantly shorter PFS than patients with low levels of these markers (AGTRAP: *p* < 0.001, log-rank; NEDD8c: *p* = 0.028, log-rank). ALKBH3, DIVERSIN, NEDD8n and RRM1 did not associate with the PFS of GBM patients, regardless if the cut-off was set according to the median-split method (Figure 5B,C,E,F) or based on the median values of the healthy tissues.

In multivariate analysis, high levels of AGTRAP significantly predicted the shorter PFS of GBM patients (HR = 1.921, [95% CI] = 1.200–3.075, *p* = 0.007). Patients with high levels of NEDD8c had an increased hazard ratio compared to NEDD8c^low^ patients, but this marker did not reach statistical significance in our cohort. The results of the multivariate analysis are summarized in Table 3.

### 3.4. Marker Combinations and the Survival of GBM Patients

In the final set of studies, we sought to determine whether our biomarker candidates would have stronger prognostic values when analysed in combination rather than individually. We only carried out these analyses for the markers that were significantly associated with OS and PFS in the univariate analysis of survival. For the analysis of OS, we selected the following test groups of patients: AGTRAP^high^ALKBH3^low^, AGTRAP^high^NEDD8c^high^, AGTRAP^high^RRM1^high^, ALKBH3^low^NEDD8c^high^, ALKBH3^low^RRM1^high^ and NEDD8c^high^RRM1^high^. Within each combination of markers, the reference group consisted of all remaining GBM patients taken together (i.e., AGTRAP^high^ALKBH3^low^ versus Rest, where ‘Rest’ included patients with AGTRAP^low^ALKBH3^low^, AGTRAP^low^ALKBH3^high^ and AGTRAP^high^ALKBH3^high^ phenotypes). The results of the multivariate analysis showed that all marker combinations were significant predictors of poor OS (Table 4). The combination of AGTRAP and ALKBH3 had the strongest prognostic value for the OS of GBM patients (HR = 2.059, 95% CI = 1.343–3.157, *p* < 0.001; Table 4). The combination of ALKBH3 and NEDD8c significantly improved the prognostic value of the individual markers (HR = 1.893, 95% CI = 1.263–2.836, *p* = 0.002; Table 4).

PFS was analysed in a similar manner. However, since only two markers were significant in the univariate analysis, there was only one possible combination in this case: AGTRAP with NEDD8c. The results showed that the AGTRAP^high^NEDD8c^high^ phenotype was a significant predictor of poor PFS in GBM patients (HR = 1.834, 95% CI = 1.123–2.995, *p* = 0.015; Table 4). 

## 4. Discussion

This study assessed the expression levels of five biomarker candidates in IDH wild-type glioblastoma (GBM) compared to healthy brain tissues and tested their prognostic values regarding overall (OS) and progression-free survival (PFS) of the patients with this type of cancer. 

For AGTRAP, we found a significantly higher expression of this marker in GBM compared to the adjacent non-malign brain parenchyma. These results are in line with a recent study by Hong et al., demonstrating that AGTRAP was overexpressed at both mRNA levels (TCGA+GTEx database) and protein levels (CPTAC database) in many tumours, including GBM [8]. Our study further showed that GBM patients with high levels of AGTRAP had a significantly shorter OS and PFS than patients with low levels of this marker. In the multivariate analysis, AGTRAP proved to be an independent prognostic factor for the OS and PFS of the GBM patients. While no detailed biomarker studies in IDH wild-type GBM were available for comparison, studies in other types of solid cancer, such as hepatocellular carcinoma [6,39], cervix carcinoma [40] and tongue squamous cell carcinoma [41], showed that AGTRAP associated with the poor outcome of those patients. Taken together, these data indicate that AGTRAP may be a valuable prognostic marker in GBM. Additionally, AGTRAP may be involved in the tumourigenesis and progression of GBM. Indeed, previous studies linked AGTRAP with the activation of well-established tumourigenic pathways such as MAPK and AKT/mTOR [6,7]. Furthermore, very recent studies found a direct correlation between AGTRAP levels and the numbers of tumour-infiltrating M2 macrophages in GBM [8]. Since M2 macrophages can drive tumour progression through immunosuppression in gliomas [42], it would be interesting for future studies to elucidate the exact involvement of AGTRAP in this context.

The role of ALKBH3 in cancer is characterised by a striking dichotomy. Previous studies in pancreatic cancer and in head and neck squamous cell carcinoma found that ALKBH3 was overexpressed in tumour compared to the healthy tissues [11,12]. In contrast, ALKBH3 levels were lower in tumour compared to healthy tissues in other types of cancer, such as breast and lung carcinomas [14,15]. In a similar manner, high levels of ALKBH3 associated with the poor outcome of patients with pancreatic cancer and renal cell carcinoma [10,12], while the opposite was observed in breast cancer, lung adenocarcinoma and Hodgkin lymphoma [13,14,15]. Recent studies by Feng et al. assessed the expression of AlkB family members, including ALKBH3, at both mRNA and protein levels in GBM. Interestingly, while the mRNA analysis using the GEPIA2 database found increased ALKBH3 levels in GBM compared to the healthy brain tissues, protein analysis using the Human Protein Atlas showed that this marker was only weakly expressed in GBM [43]. The latter data are in line with our study, which found significantly lower protein levels of ALKBH3 in GBM compared to the healthy brain parenchyma, as well as an association of the ALKBH3^low^ phenotype with a short OS of the GBM patients. A possible explanation for the discrepancy between mRNA and protein data is the relatively high instability of the ALKBH3 protein due to proteasome targeting. This is supported by comprehensive biochemical studies on ALKBH3 ubiquitination, which found that ALKBH3 mRNA levels did not correlate well with ALKBH3 protein expression in prostate, breast or lung carcinoma cell lines [44]. While the mechanisms of ALKBH3 ubiquitination in GBM still require characterisation, it is not excluded that a similar phenomenon occurs in this type of cancer as well. 

DIVERSIN/ANKRD6 is a tumour-promoting factor, which has been proposed as prognostic biomarker in several types of cancer. Specifically, DIVERSIN was found to be overexpressed in tumours compared to healthy adjacent tissues in colorectal carcinoma, non-small cell lung cancer and breast cancer [17,18,20]. For the same tumour entities, patients with high levels of DIVERSIN had a significantly worse outcome than patients with low levels of this marker [17,18,20,21]. Our study did not find an association between DIVERSIN expression and the OS and/or PFS of GBM patients. However, we did find significantly higher levels of DIVERSIN in GBM compared to non-malign brain tissues. These results are supported by previous studies by Wang et al., showing that DIVERSIN was overexpressed in glioma tissues and positively correlated with the WHO grade of the gliomas [19]. Using siRNA knockdown, the authors additionally showed that DIVERSIN was involved in the proliferation and invasion of glioma cell lines. Together, these data indicate that DIVERSIN—while not suitable as a prognostic marker—may be nevertheless a tumourigenic factor and, thus, a potential therapeutic target in GBM.

Accumulating evidence indicates that neddylation plays important roles in the pathophysiology and therapy of cancer (reviewed in [23,24]). This mechanism received particular attention in recent years, since the neddylation inhibitor MLN4924/pevonedistat showed promising anti-tumour effects in phase I–III clinical trials [24]. It is therefore surprising that only few studies addressed the prognostic relevance of NEDD8 expression for the outcome of cancer patients thus far. Specifically, Tian et al. showed that bladder carcinoma patients with high NEDD8 tumour levels had a significantly shorter OS and PFS compared to their NEDD8^low^ counterparts [45]. Similar results were obtained by Xian et al. in esophageal squamous cell carcinoma [46]. In gliomas, Hua et al. found that NEDD8 expression correlated with the WHO grade, and that high NEDD8 levels associated with a shorter OS in WHO grade 4 patients [47]. Our own results are in line with these findings. However, it should be pointed out that the scoring system used in previous studies did not take into consideration the subcellular localisation of NEDD8. In the present study, we observed that NEDD8 had both a cytoplasmic and a nuclear distribution. We therefore assessed the expression levels of NEDD8 in the two cellular compartments separately (NEDD8c versus NEDD8n). Interestingly, we found that only NEDD8c was overexpressed in GBM tissues compared to the healthy brain. Furthermore, high levels of NEDD8c significantly associated with a shorter OS and PFS of the GBM patients. In contrast, NEDD8n was equally expressed in tumour and healthy tissues and did not associate with the patients’ outcome. These findings suggest that the tumour-promoting functions of NEDD8 may also be a consequence of its cytoplasmic localisation. Although no direct evidence is currently available to support this hypothesis, previous studies found that NEDD8 could translocate from the nucleus to the cytoplasm in response to Interleukin-1β [48]. Since this cytokine is known to promote GBM proliferation, invasion and angiogenesis (reviewed in [49]), it would be tempting to speculate that an Interleukin-1β/NEDD8 signaling axis is also involved in the progression of GBM.

The last marker tested in our study was RRM1. This protein was proposed as a prognostic factor for OS in gastric cancer [28], pancreatic cancer [27] and biliary tract cancer [26]. A meta-analysis of 23 studies on non-small cell lung cancer found that high levels of RRM1 significantly associated with a shorter OS of the patients, though the prognostic significance was not reached in Cox regression analysis [29]. Using the CGGA and Rembrandt databases, Jiang et al. found that the mRNA levels of RRM1 were increased in high-grade gliomas (WHO grade 3 and 4) compared to healthy brain tissues. Within the WHO grade 4 group, there was a significant association between high RRM1 mRNA expression and poor OS of these patients [31]. Very recently, Ariey-Bonnet et al. elegantly demonstrated that RRM1 was a top target for drug combination therapy in GBM. The authors additionally evaluated RRM1 expression by immunohistochemistry in a cohort of 97 GBM patients and showed that high levels of RRM1 predicted poor OS of the patients in a Cox regression model adjusted by type of surgery [32]. Our study found that GBM tissues had significantly higher levels of RRM1 compared to the healthy brain tissues. When the cut-off was set according to the baseline expression of the healthy tissues, the RRM1^high^ phenotype was significantly associated with the poor OS of the GBM patients. Furthermore, we identified RRM1 as a significant prognostic factor for the OS of GBM patients, which was independent of age, Karnofsky performance scale, extent of surgical resection and MGMT methylation status. Taken together, these data indicate that RRM1 may be a valuable biomarker and, importantly, a promising therapeutic target in GBM.

Previous studies showed that combined analysis of multiple markers had superior prognostic value compared to the analysis of individual markers [50,51,52,53]. Here, we performed two-by-two combinations of the markers that significantly associated with the OS (AGTRAP, ALKBH3, NEDD8c and RRM1) or the PFS (AGTRAP and NEDD8c) of GBM patients in univariate survival analysis. We found that all combinations of markers were significant predictors for the OS or PFS in these patients. However, only two combinations (AGTRAP/ALKBH3 and ALKBH3/NEDD8c) had superior prognostic value compared to the individual markers. The other four combinations did not have significantly increased prognostic values compared to AGTRAP or RRM1 analysed individually. An overview of the prognostic accuracy of the markers evaluated in this study is provided in Figure 6.

## 5. Conclusions

This study identifies novel individual and combination markers with prognostic relevance for the outcome of IDH wild-type GBM patients. Our results indicate that AGTRAP and RRM1 are prognostic markers for the OS, while AGTRAP is also a prognostic marker for the PFS of these patients. Combining AGTRAP with ALKBH3 may improve the prognostic accuracy for the OS of the GBM patients. Since ALKBH3 and NEDD8c did not reach prognostic significance when analysed individually, their combination could be useful to predict the patients’ OS more accurately. Together with our studies on tumour versus non-malign brain tissues, these findings contribute to a better understanding of GBM pathophysiology and may help identify future targets for improved therapeutic strategies in this type of cancer.

## Figures and Tables

**Figure 1 biomedicines-12-00926-f001:**
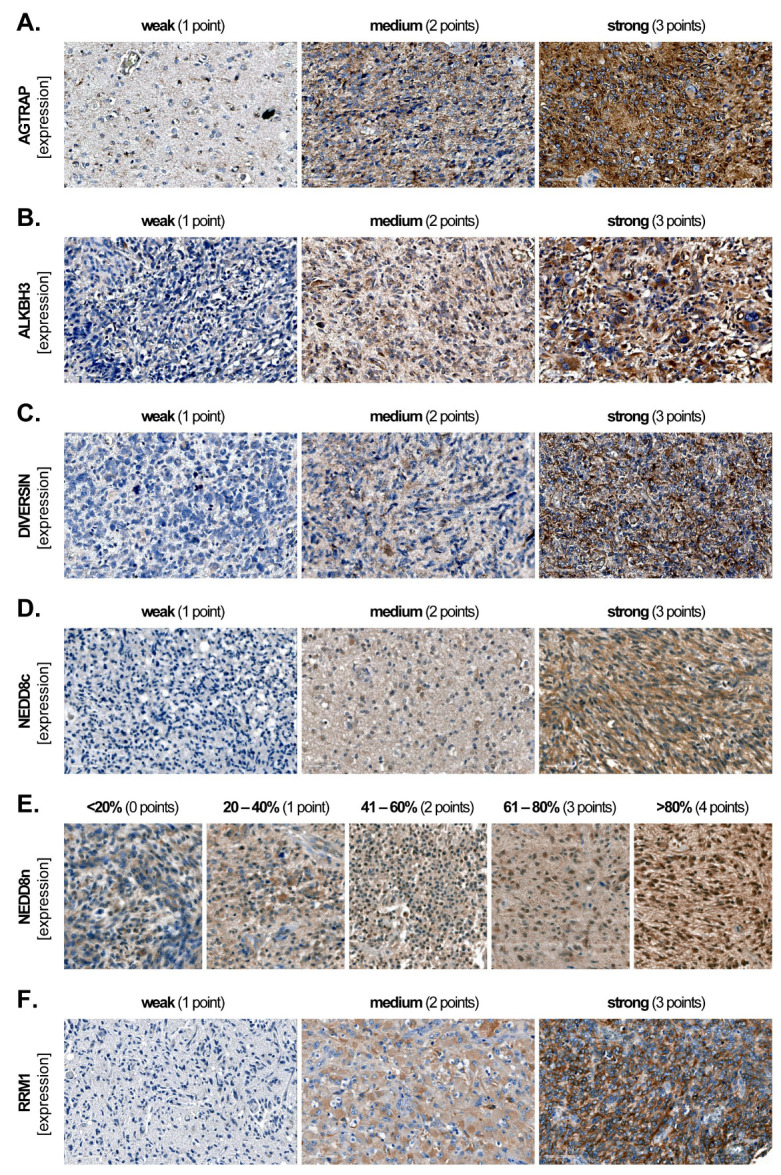
**Marker expression in GBM tissues.** Representative micrographs showing weak (1 point), medium (2 points) and strong (3 points) cytoplasmic expression of (**A**) AGTRAP, (**B**) ALKBH3, (**C**) DIVERSIN, (**D**) NEDD8c and (**F**) RRM1. The H-score was subsequently calculated according to the formula (1 × X) + (2 × Y) + (3 × Z), where X + Y + Z = 100% of the total tumour area. (**E**) The 5-tier score for nuclear NEDD8 (NEDD8n) according to the percentage of positive cells.

**Figure 2 biomedicines-12-00926-f002:**
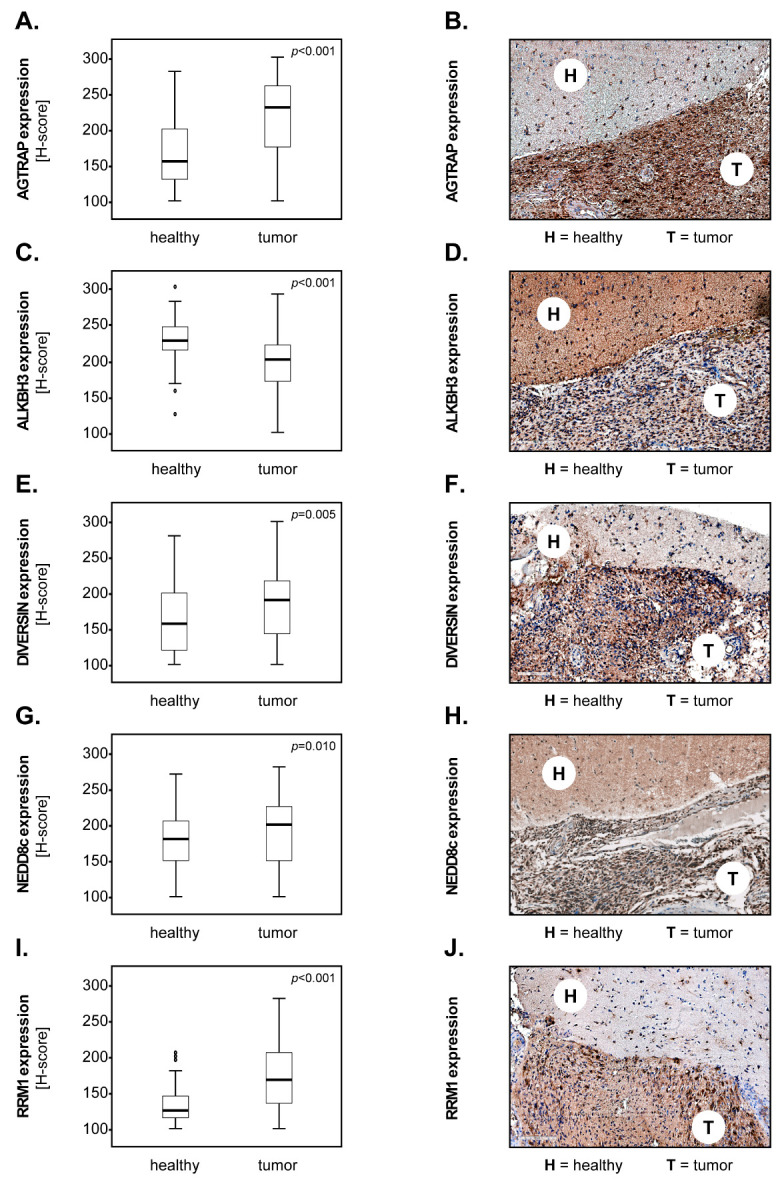
**Marker expression in healthy versus GBM tissues.** Expression of (**A**) AGTRAP, (**C**) ALKBH3, (**E**) DIVERSIN, (**G**) NEDD8c and (**I**) RRM1 in GBM (n = 186) and tumour-free adjacent brain tissues (n = 54). The medians are shown as black lines and the percentiles (25th and 75th) as vertical boxes with error bars. The outliers are indicated by circles. Statistical analysis was performed with the Mann–Whitney U test, and the *p*-values are indicated in the upper-right corner of each plot. (**B**,**D**,**F**,**H**,**J**) Representative micrographs showing the expression of the markers in the solid tumour area (T) versus the adjacent, tumour-free tissue area (H).

**Figure 3 biomedicines-12-00926-f003:**
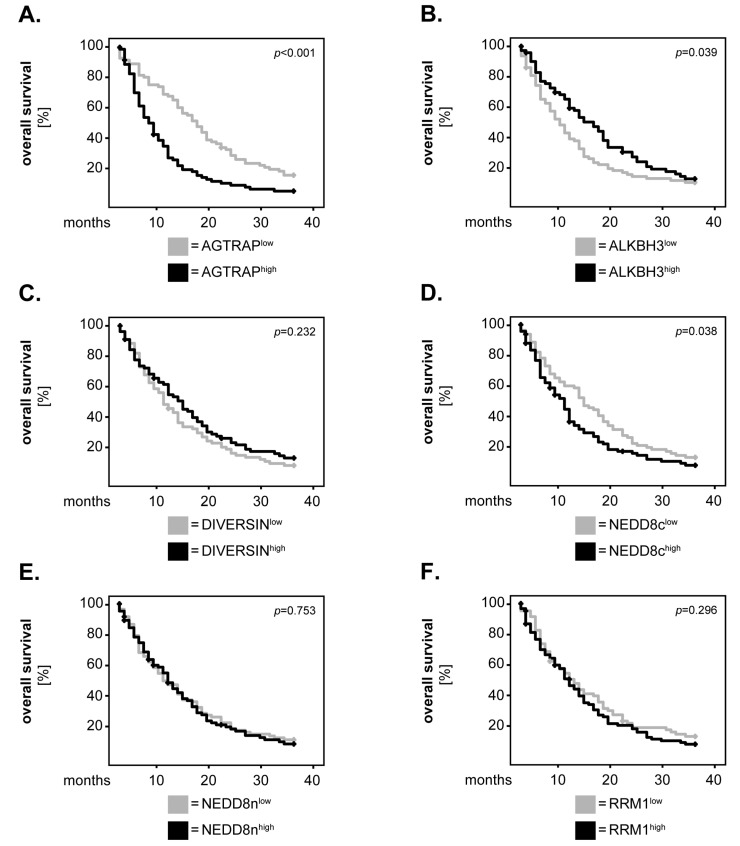
**Marker expression and the overall survival of GBM patients—univariate analysis.** (**A**–**F**) The expression levels of the markers were dichotomised into ‘low’ and ‘high’ according to the median-split method. Kaplan–Meier curves were generated for the 36-month overall survival, and statistical analysis was performed with the log-rank test. The *p*-values are indicated in the upper-right corner of each plot.

**Figure 4 biomedicines-12-00926-f004:**
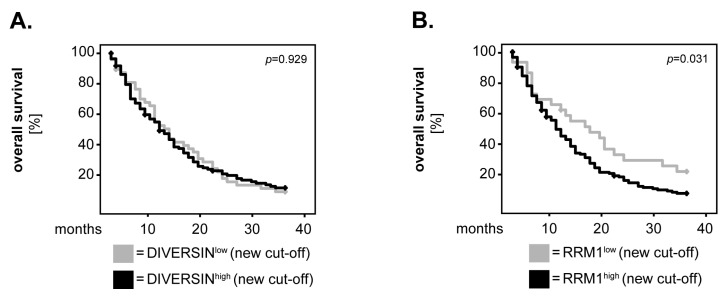
**New univariate analysis of DIVERSIN and RRM1 in relation to the overall survival of GBM patients.** The expression levels of (**A**) DIVERSIN and (**B**) RRM1 were dichotomised into ‘low’ and ‘high’ according to the median values of the healthy tissues. Kaplan–Meier curves were generated for the 36-month overall survival, and statistical analysis was performed with the log-rank test. The *p*-values are indicated in the upper-right corner of each plot.

**Figure 5 biomedicines-12-00926-f005:**
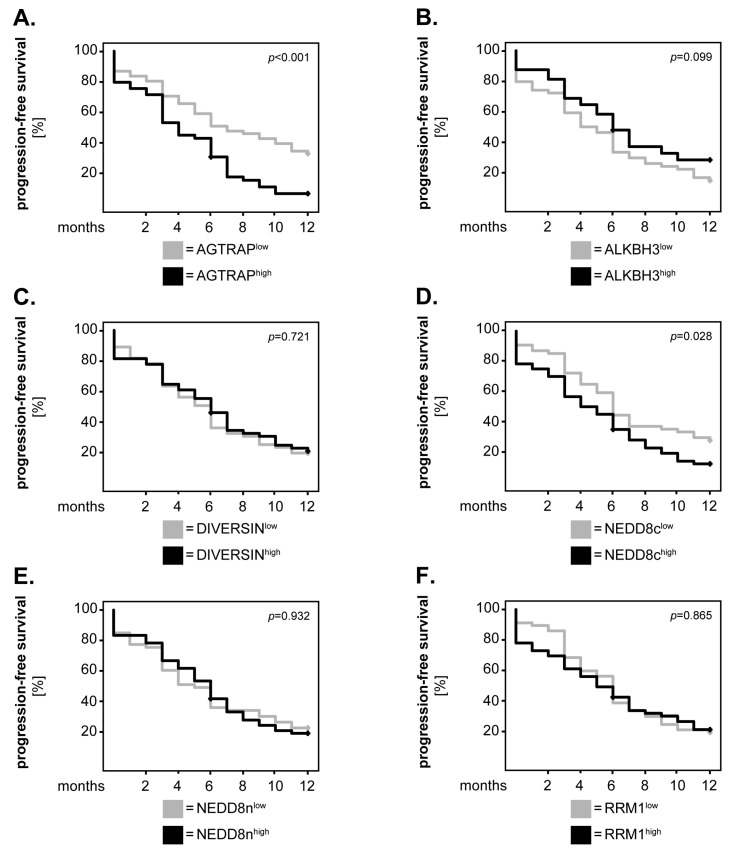
**Marker expression and the progression-free survival of GBM patients—univariate analysis.** (**A**–**F**) The expression levels of the markers were dichotomised into ‘low’ and ‘high’ according to the median-split method. Kaplan–Meier curves were generated for the 12-month progression-free survival and statistical analysis was performed with the log-rank test. The *p*-values are indicated in the upper-right corner of each plot.

**Figure 6 biomedicines-12-00926-f006:**
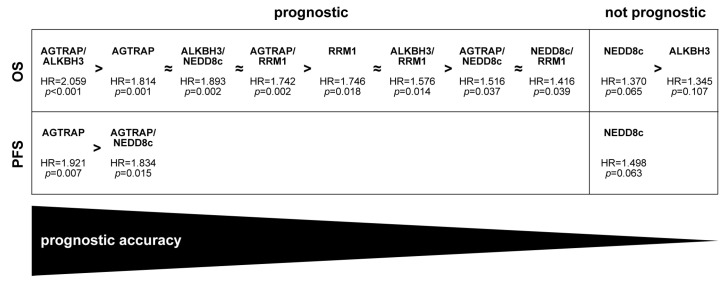
**Prognostic accuracy of individual and combination markers regarding OS and PFS of GBM patients.** Only the markers that reached significance in the univariate survival analysis are included in this overview. Markers with stronger prognostic value are indicated by ‘>’. Markers with approximately equal prognostic value are indicated by ‘≈’.

**Table 1 biomedicines-12-00926-t001:** Clinical characteristics of the GBM patients included in this study. RTX: radiotherapy; CTX: chemotherapy; RCTX: radiochemotherapy; n.d.: not determinable.

Clinical Parameter	Number	Percentage
**Sex**		
female	85	45.7
male	101	54.3
**Karnofsky performance scale**		
10	3	1.6
20	2	1.1
30	11	5.9
40	8	4.3
50	29	15.6
60	33	17.7
70	60	32.3
80	19	10.2
90	19	10.2
100	0	0
n.d.	2	1.1
**Therapy**		
surgery	35	18.8
surgery+RTX	21	11.3
surgery+CTX	1	0.5
surgery+RCTX	123	66.1
n.d.	6	3.2
**Extent of surgical resection**		
total	65	34.9
subtotal	111	59.7
n.d.	10	5.4
**MGMT methylation status**		
unmethylated	70	37.6
methylated	116	62.4
n.d.	0	0
**IDH mutation status**		
wild-type	186	100
mutated	0	0
n.d.	0	0

**Table 2 biomedicines-12-00926-t002:** Multivariate Cox regression analysis of individual markers in relation to overall survival. The reference variable (dummy) is indicated by HR = 1. The model was adjusted for age, Karnofsky performance scale, surgical resection and MGMT methylation status. HR: Hazard Ratio; CI [95%]: 95% confidence interval.

Cox Regression: OS	HR	CI [95%]	*p*-Value
AGTRAP^low^	1		
AGTRAP^high^	1.814	1.260–2.612	0.001
ALKBH3^high^	1		
ALKBH3^low^	1.345	0.938–1.929	0.107
NEDD8c^low^	1		
NEDD8c^high^	1.370	0.980–1.914	0.065
RRM1^low^ (new cut-off)	1		
RRM1^high^ (new cut-off)	1.746	1.099–2.775	0.018

**Table 3 biomedicines-12-00926-t003:** Multivariate Cox regression analysis of individual markers in relation to progression-free survival. The reference variable (dummy) is indicated by HR = 1. The model was adjusted for age, Karnofsky performance scale, surgical resection and MGMT methylation status. HR: Hazard Ratio; CI [95%]: 95% confidence interval.

Cox Regression: PFS	HR	CI [95%]	*p*-Value
AGTRAP^low^	1		
AGTRAP^high^	1.921	1.200–3.075	0.007
NEDD8c^low^	1		
NEDD8c^high^	1.498	0.979–2.293	0.063

**Table 4 biomedicines-12-00926-t004:** Multivariate Cox regression analysis of combined markers in relation to overall and progression-free survival. The reference variable (dummy) is indicated by HR = 1. The model was adjusted for age, Karnofsky performance scale, surgical resection and MGMT methylation status. HR: Hazard Ratio; CI [95%]: 95% confidence interval.

Cox Regression: OS	HR	CI [95%]	*p*-Value
Rest	1		
AGTRAP^high^ALKBH3^low^	2.059	1.343–3.157	<0.001
AGTRAP^high^NEDD8c^high^	1.516	1.025–2.243	0.037
AGTRAP^high^RRM1^high^	1.742	1.223–2.480	0.002
ALKBH3^low^NEDD8c^high^	1.893	1.263–2.836	0.002
ALKBH3^low^RRM1^high^	1.576	1.096–2.266	0.014
NEDD8c^high^RRM1^high^	1.416	1.017–1.971	0.039
**Cox regression: PFS**	**HR**	**CI [95%]**	***p*-value**
Rest	1		
AGTRAP^high^NEDD8c^high^	1.834	1.123–2.995	0.015

## Data Availability

The datasets on patient material are not publicly available as they also contain individual participants’ details. These data can be obtained from the corresponding author in anonymised form upon reasonable request.

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
