# Peer review of "The Roles of AGTRAP, ALKBH3, DIVERSIN, NEDD8 and RRM1 in Glioblastoma Pathophysiology and Prognosis"

_biomedicines, 2024, doi:10.3390/biomedicines12040926_

Round 1
Reviewer 1 Report
Comments and Suggestions for Authors
The manuscript entitled "The roles of AGTRAP, ALKBH3, DIVERSIN, NEDD8 and 2 RRM1 in glioblastoma pathophysiology and prognosis" and submitted to Biomedicines is aimed to study the novel biomarkers of IDH wild-type glioblastoma (GBM) tissues compared to non-malign brain parenchyma, as well as their prognostic relevance for the GBM patients’ outcome. The manuscript is well written, the research carried out is characterized by a well-planned design, well-conducted experiments and clear results and statistical analysis. I believe that the manuscript can be published in its current form.
Comments on the Quality of English LanguageMinor editing of English language required.
Author Response
General remarks:
- main changes and additions in the manuscript text are highlighted in yellow.
- two new supplementary figures (Figure S1 and Figure S2 A-E) were included in the revised version of the manuscript.
Comment: The manuscript is well written, the research carried out is characterized by a well planned design, well-conducted experiments and clear results and statistical analysis. I believe that the manuscript can be published in its current form. Minor editing of English language required.
Reply: We thank the reviewer for the positive evaluation of our study. According to the reviewer’s suggestion, we thoroughly checked the manuscript for grammar, spelling and syntax.
Reviewer 2 Report
Comments and Suggestions for Authors
The research article by Dr. Dumitru and colleagues, titled "The Roles of AGTRAP, ALKBH3, DIVERSIN, NEDD8, and RRM1 in Glioblastoma Pathophysiology and Prognosis," utilized immunohistochemistry to assess the expression of five glioblastoma (GBM) biomarker candidates in n=186 tumor tissues compared to n=54 healthy brain tissues. The study also investigated their association with patients’ overall survival (OS) and progression-free survival (PFS), as well as their prognostic relevance.
The authors identified that four of these novel biomarkers, AGTRAP, cytosolic NEDD8 (NEDD8c), and RRM1 ALKBH3, are overexpressed in GBM brain tumors compared to healthy brain tissue, while ALKBH3 is highly expressed in healthy brain tissue but lowly expressed in tumors. Furthermore, AGTRAP, ALKBH3, NEDD8c, and RRM1 are significantly associated with OS, with AGTRAP and RRM1 also acting as independent prognostic factors. Additionally, AGTRAP and NEDD8c are significantly associated with PFS, with AGTRAP being an independent prognostic factor for PFS. The combination of AGTRAP/ALKBH3 was found to possess the strongest prognostic value for the OS of GBM patients.
Although this study contributes to a better understanding of GBM pathophysiology and aids in planning new therapies, it has limitations that must be considered to avoid overinterpretation of the results. The novel biomarkers were solely analyzed through immunohistochemistry, without quantitative analysis or expression level documentation for each protein. This lack of quantitative data makes it challenging to establish solid correlations between these markers and OS/PFS in GBM patients, limiting their prognostic value. While the authors refer to expression level analyses conducted by other studies in the discussion, this does not compensate for the absence of quantitative values in the present study.
Furthermore, the significant difference in sample sizes between tumor tissues (n=186) and non-malignant brain parenchyma (n=54) weakens any comparative analysis. The heterogeneity in cellular composition of the analyzed tissues also needs consideration when interpreting the positivity of samples for each marker, as immunopositivity was established regardless of the specific cell type (vascular, glial, neuronal).
The authors' attempt to assign quantitative values to the immunohistochemical signal based on intensity and extension of positive areas is noted, but it remains challenging to discern differences between points assigned to images, such as those shown in Figure 1E. Additionally, the clear delimitation seen in images of GBM and adjacent healthy tissue in Figure 2 is unusual, as GBM tumors typically infiltrate normal tissue without clear borders. This discrepancy raises questions about the representativeness of the images for the majority of tumor samples and warrants further investigation.
Author Response
see attached PDF file

Round 2
Reviewer 2 Report
Comments and Suggestions for Authors
The authors provided insightful replies to all points raised by this reviewer.